# Barriers and Facilitators of Availability of Hydroxyurea for Sickle Cell Disease in Tanzania; A Qualitative Study of Pharmaceutical Manufacturers, Importers, and Regulators

**DOI:** 10.3390/healthcare10112223

**Published:** 2022-11-07

**Authors:** Hamu J. Mlyuka, Manase Kilonzi, Ritah F. Mutagonda, Lulu Chirande, Wigilya P. Mikomangwa, David T. Myemba, Godfrey Sambayi, Dorkasi L. Mwakawanga, Joyce Ndunguru, Agnes Jonathan, Julie Makani, Paschal Ruggajo, Irene K. Minja, Emmanuel Balandya, Appolinary A. R. Kamuhabwa

**Affiliations:** 1Department of Clinical Pharmacy and Pharmacology, School of Pharmacy, Muhimbili University of Health and Allied Sciences, Dar es Salaam P.O. Box 65013, Tanzania; 2Sickle Pan African Research Consortium (SPARCO), Dar es Salaam P.O. Box 65001, Tanzania; 3Department of Paediatrics and Child Health, Muhimbili University of Health and Allied Sciences, Dar es Salaam P.O. Box 65001, Tanzania; 4Department of Pharmaceutics and Pharmacy Practice, School of Pharmacy, Muhimbili University of Health and Allied Sciences, Dar es Salaam P.O. Box 65013, Tanzania; 5Department of Pharmacognosy, School of Pharmacy, Muhimbili University of Health and Allied Sciences, Dar es Salaam P.O. Box 65013, Tanzania; 6Department of Community Health Nursing, School of Nursing, Muhimbili University of Health and Allied Sciences, Dar es Salaam P.O. Box 65001, Tanzania; 7Sickle Cell Program, Department of Hematology and Blood Transfusion, School of Medicine, Muhimbili University of Health and Allied Sciences, Dar es Salaam P.O. Box 65001, Tanzania; 8Nephrology Unit, Department of Internal Medicine, School of Medicine, Muhimbili University of Health and Allied Sciences, Dar es Salaam P.O. Box 65001, Tanzania; 9Department of Restorative Dentistry, School of Dentistry, Muhimbili University of Health and Allied Sciences, Dar es Salaam P.O. Box 65001, Tanzania; 10Department of Physiology, School of Medicine, Muhimbili University of Health and Allied Sciences, Dar es Salaam P.O. Box 65001, Tanzania

**Keywords:** pharmaceutical importers and manufacturers, sickle cell disease, hydroxyurea, barriers and facilitators, Tanzania

## Abstract

Despite three decades of proven safety and effectiveness of hydroxyurea in modifying sickle cell disease (SCD), its accessibility is limited in Sub-Saharan Africa, which shares 75% of the world’s SCD burden. Therefore, it is time to explore the barriers and facilitators for manufacturing and importation of hydroxyurea for SCD in Tanzania. This was qualitative research that employed a case study approach. Purposive sampling followed by an in-depth interview (IDI) using a semi-structured questionnaire aspired by data saturation enabled us to gather data from 10 participants. The study participants were people with more than three years of experience in pharmaceuticals importation, manufacturing, and regulation. The audio-recorded data were verbatim transcribed and analyzed using thematic analysis. Two themes were generated. The first comprised barriers for importation and manufacturing of hydroxyurea with sub-themes such as inadequate awareness of SCD and hydroxyurea, limited market, and investment viability. The second comprised opportunities for importation and manufacturing of hydroxyurea with sub-themes such as awareness of activities performed by medicines regulatory authority and basic knowledge on SCD and hydroxyurea. Inadequate understanding of SCD, hydroxyurea, and orphan drug regulation are major issues that aggravate the concern for limited market and investment viability. Existing opportunities are a starting point towards increasing the availability of hydroxyurea.

## 1. Introduction

Sickle cell disease (SCD) is an inherited hemoglobinopathy characterized by a defect in the β-globin chain of hemoglobin of the red blood cells (RBC). The ensuing susceptibility to hemolysis and vaso-occlusion are the hallmarks of SCD and may lead to severe anemia, severe pain, stroke, and multiple organ damage [1]. The World Health Organization (WHO) and United Nations (UN) have recognized SCD as a global public health problem [1]. Every year, 300,000 babies are born with SCD worldwide [2]. Sub-Saharan Africa contributes 75% of the global burden of SCD [2,3]. Makani et al. (2011 and 2015) reported that, in Tanzania, more than 11,000 children are born with SCD every year [2,3].

The detrimental health effects of SCD include high morbidity and mortality [1,4]. While in developed countries, a significant proportion of SCD patients reach their 50th birthday, in Africa, more than 50% of patients with SCD do not celebrate their 20th birthday [1]. Efforts to overcome the suffering and premature death due to SCD have yielded several practical interventions, namely blood transfusion, prophylactic antibiotics, vaccines, disease-modifying agents, and bone marrow transplant. Taking a case of Tanzania, there have been great strides made since the 1980s, when the first SCD clinic was opened at Muhimbili National Hospital [5]. For the past two decades, the focus was on establishing comprehensive care, whereby government has been working with different stakeholders such as Muhimbili University of Health and Allied Sciences (MUHAS) Sickle Cell Program (SCP) and civil society organizations. Advocacy programs aiming at SCD awareness creation are mostly championed by civil society organizations such as Sickle Cell Youth Foundation, Tanzania Sickle Cell Disease Alliance, Sickle Cell Foundation of Tanzania, and Sickle Cell Disease Patients Community [5]. MUHAS SCP started in 2004 with a focus on research for improvement of SCD care, advocacy, and training. Thus far, MUHAS SCP has a cohort of 5466 individuals with SCD and is leading research on newborn screening [6,7] and curative care (hematopoietic stem cells and gene therapy) [5]. In addition, through collaboration, MUHAS SCP and the Tanzania ministry of health have managed to develop SCD management guidelines useful [5] at all three levels of the healthcare system, namely primary, secondary, and tertiary [8]. Primary and secondary healthcare facilities provide basic care, while the tertiary level provides more advanced comprehensive care for SCD. Therefore, the majority of people with SCD receives their healthcare in tertiary hospitals. Generally, the current recommended interventions for SCD in Tanzania are oral phenoxy-methyl penicillin/penicillin V and immunization against *Streptococcus pneumoniae* using the pneumococcal conjugate vaccine (PCV-13) [9], blood transfusion, and hydroxyurea. 

Hydroxyurea is an SCD-modifying agent that increases production of fetal hemoglobin, which is not prone to precipitation in a hypoxemic condition [1,10]. The drug has been in use for more than 30 years and has proven to be safe and effective [11]. The drug has been proven to reduce morbidity and mortality as well as improve the quality of life among patients with SCD. In countries such as the USA, hydroxyurea is initiated to every child immediately after being diagnosed with SCD [12,13].

Paradoxically, despite the proven benefits of hydroxyurea, there is poor utilization worldwide [14]. The problem is even worse in Sub-Saharan Africa regardless of its contribution to the global burden of SCD. One of the factors for underutilization of hydroxyurea is its limited availability and accessibility [3]. The rare nature of the SCD has been cited as the reason for underinvestment by pharmaceutical manufactures in different countries [12,15]. This has resulted in the scarcity of the hydroxyurea and its high price, jeopardizing its accessibility to the needy [14].

Some of the steps taken by different countries to address the concern of the limited investment in life-saving medications for rare diseases such as SCD include development of different policy and regulation [15]. The aim is to provide favorable environment for pharmaceutical companies to invest in rare diseases [16]. Similar steps have been taken by Tanzania government, which includes some tax exemption to pharmaceutical companies, and Tanzania Medicines and Medical Devices (TMDA)’s orphan drug regulation of 2018 [17]. However, despite the presence of these favorable policies and regulations, there is no pharmaceutical manufacturer that produces hydroxyurea in Tanzania. In addition, there is only one registered brand of hydroxyurea imported to Tanzania. Therefore, with this problem in hand, we decided to explore contemporary barriers and prospective facilitators for manufacturing and importation of hydroxyurea for SCD in Tanzania. 

## 2. Materials and Methods

### 2.1. Study Design and Settings

A qualitative study design [18] was used to explore pharmaceutical experts’ experiences with and perception of manufacturing and importation and regulation of hydroxyurea in Tanzania. It was conducted in Dar es Salaam, a region that hosts the sub-headquarter of the medicines regulatory authority, all pharmaceutical importers, and five out of six manufacturers in the country. Another region was Mwanza, where there is one pharmaceutical manufacturing industry.

### 2.2. Participants and Recruitment

The study employed a case study approach, whereby participants with unique characteristics with respect to pharmaceuticals importation, manufacturing, and regulation were purposively sampled. All study participants were supposed to have a working experience of more than three years. We recruited superintendent pharmacists working in pharmaceutical importers, which are currently the marketing authorization holder of at least one product used in management of anemia, such as folic acid or iron supplements. For manufacturers, we recruited head industrial pharmacists working in pharmaceutical industries whose portfolio includes products of liquid- and solid-dosage formulation. The eligibility criteria of importers were assessed through the TMDA website section on registered medicines. In total, six importers and five manufacturers were eligible for this study. With respect to pharmaceutical regulators, we recruited those who are involved with medicines registration, specifically orphan drugs. Failure to consent was considered as exclusion criteria. 

### 2.3. Data Collection

#### 2.3.1. Preliminary Procedures

After identification of eligible stakeholders, the team, through the principal investigator (PI), wrote request permission to conduct research in a respective company or institution. The letter stated who qualified for this study. It was attached with the research proposal and ethical clearance letter for the approval of this study. One importer and one manufacturer of pharmaceuticals were found to be defunct, while one pharmaceuticals importer declined to participate. After receiving feedback, the research team made arrangement for the interview. 

#### 2.3.2. Data Collection Method

This study employed an in-depth interview using a semi-structured interview guide to collect data from study participants. The research team developed the preliminary semi-structured interview guide with open-ended questions and probes to explore and understand better the issues of relevancy on the use of hydroxyurea in managing SCD and its manufacturing and importation. As a matter of flexibility, the semi-structured guide was improved regularly to respond to the emergent ideas so as to gain greater amounts of deep insight in the subsequent interviews.

#### 2.3.3. In-Depth Interviews

Two researchers (H.J.M. and M.K.) were responsible for conducting in-depth interviews. Interviews were conducted in a room chosen by the participant to ensure comfort and privacy in the buildings of the respective institution. Each interview was conducted at the time of convenience by the study participant based on a pre-arranged appointment. The interview commenced after the participant had been briefed about the purpose of the study, and he/she had signed the consent form. All interviews were conducted in the Kiswahili language, as it is a native language of all participants and authors. Interviews were audio-recorded to capture the information provided by the participants. Additionally, researchers took field notes on verbal and non-verbal responses to compliment the recorded audio information. Each interview lasted between 20 and 30 min. Immediately after the end of the interview, the moderator or lead interviewer asked the interviewee to clarify some recorded notes or provide mentioned documents to enrich what he/she had said. We reviewed the field notes on daily basis to improve the subsequent interviews and to note the emerging findings. It was at the 7th interview that we realized the reoccurrence of similar themes with few new findings. At the 10th interview, we decided to stop further interviews, as there was no new information being uncovered from the interviews. Therefore, we considered the 10th interview as our point of information saturation. 

### 2.4. Data Management

All collected data in electronic and hardcopies were kept on a computer and in a locked case accessed by the principal investigator only. 

### 2.5. Data Analysis

This study employed inductive thematic analysis, as documented in these articles [19,20], owing to its flexibility and ability to accommodate researchers with different backgrounds. Initial data analysis was conducted by six researchers (H.J.M., M.K., D.L.M., D.T.M., W.P.M., and G.S.), and refinement of the findings involved all authors. Except for D.L.M., who is a midwife and qualitative researcher, the rest are pharmacists with specializations in clinical pharmacy (H.J.M.), pharmacology and therapeutics (M.K. and W.P.M.), and industrial pharmacy (D.T.M.). The process of data analysis started with verbatim transcription of the audio records. The transcription was conducted by expert transcriber in collaboration with the principal investigator. We had two data sets: the first from transcripts and the other from field notes. We decided to use all the data sets for the sake of having a rich data source. The subsequent steps of data analysis were in a cyclic process and an iterative one, but we loosely grouped them into five steps. The first step was familiarization with data, whereby all data-analyzing team members were given transcripts and field notes to read and re-read. This was followed by the development of initial codes, which involved deduction from existing theories and induction from emergent ideas noted during the familiarization step. The third step was open-coding, whereby each transcript was coded by two researchers, and since the process took place in the meeting hall, it allowed for consensus through group discussion whenever ambiguity arose during coding so as to reduce inter-code variability. New emergent codes were added in the code book. After coding, the fourth step was combining codes, whereby the research team observed for commonalities and differences to develop sub-themes. Lastly, the team abstracted sub-themes to develop overarching themes. The refinement of the findings involved all authors with different research backgrounds in medicine, pharmacy, and nursing. 

### 2.6. Ethical Considerations 

The ethical approval to conduct this study was obtained from the MUHAS research and ethics committee (Ref. No. MUHAS-REC-07-2020-302). Permissions to conduct the study were obtained from each respective institution where study participants were working. Ethical standards were upheld during data collection, whereby all participants were well informed about the study, audio recording, and their right to decide to participate or refrain at any stage. Interviews commenced after the participants signed an informed consent form. To increase confidentiality, no name of any individual was mentioned during the interview, and numbers were used to identify participants. 

## 3. Results

### 3.1. Study Participants

A total of 10 participants was interviewed, as shown in Table 1 below. The majority of the study participants were male, with a degree level of education, and were pharmacist by profession. Experience ranged from 5 to 56 years.

### 3.2. Emergent Themes

This study generated two themes, which are barriers to importation and manufacturing of hydroxyurea and opportunities that may facilitate importation and manufacturing of hydroxyurea (Table 2 below).

#### 3.2.1. Barriers to Production and Importation of Hydroxyurea

This theme was abstracted from four sub-themes, namely inadequate awareness of sickle cell disease and hydroxyurea, limited market and investment viability, lack of infrastructure and resources to import or manufacture hydroxyurea, and inadequate knowledge on orphan diseases, medicines, and the Tanzania Orphan Drug Regulation of 2018.

##### Inadequate Awareness of Sickle Cell Disease and Hydroxyurea

Participants lacked actual data on the burden of SCD in Tanzania and how hydroxyurea is used in the management of SCD. For instance, one manufacturer participant said “…we don’t know who needs the treatment (hydroxyurea) and what does the treatment require, how long you have to give…”. M01 

Similarly, many importers were puzzled by the issue of the burden of SCD. One participant gave the following statement:

“…if the drug is used specifically for the treatment of sickle cell disease, how many patients do we have in the country? Those who are under treatment and is it only hydroxyurea or there are other alternatives which can be used in the treatment of sickle cell disease…”.I04

Lack of understanding about the burden of sickle cell and use of hydroxyurea to these important stakeholders was also perceived by a regulatory officer who said that the lack of data on the burden of disease hinders manufacturers or importers from harnessing the incentives offered in the Tanzania Orphan Drug Regulation of 2018. A quoted example includes the following:

“Majority have the wish, want to register a product as an orphan medicine but they face difficulties to prove that this drug treats a disease which affects few people”.R01

Apart from the failure to generate research evidence on the burden of sickle cell disease, some of the participants were not aware of how to generate those data. For example, one participant said, “…I don’t have any mechanism to prove that this product is less used by the society so it falls under orphan drugs…”. I01

This was the reason why one regulatory officer insisted on inter-institutional collaboration to make sure data on disease burden are easily accessible through the ministry responsible for health.

##### Limited Market and Investment Viability

This was the most reported barrier by participants from manufacturers to importers regardless of their understanding of the burden of SCD. From a business perspective, manufacturers and importers think that SCD has a limited market, and doing business with this niche is not lucrative. 

“So, you find that if the disease, the sickle cell is not just an emergency to the community and therefore the demand for the products which are used to treat sickle cell anemia is not of that much attention to the people, so the manufacturer will not opt for that product to be manufactured because it may be difficult for them to sell”.M03

“The main reason first was, because we are more business-oriented and we saw its market (of hydroxyurea) is so limited”. M02

“The first criteria we look on is the market availability, or the matching of market needs…”. M04

Moreover, the limited market for hydroxyurea instigated fear for many importers, whereby they think it is a profitless business that has a high risk of product loss due to stock expiration since it will be a slow-moving product. One importer said the following:

“…Indeed, like us importers because we are dealing with importation and distribution, always you want to import product which will give you profit meaning that it has costumers, it has a market. So you can’t import product which has no or has few customers”.I04

Another importer make the following statement: 

“Key barriers, the first barrier is limited demand…because we do not have any demand for this product, so if there is no demand then who will buy from me if I import that (hydroxyurea) product”?I01

Recommendations to overcome the limited market and investment viability included a request for the government to subsidize raw materials for hydroxyurea manufacturing or take responsibility to cover for operational cost. Furthermore, one manufacturer asked for partnership with World Health Organization (WHO) and United Nations Children’s Emergency Fund (UNICEF) to support the manufacturing of hydroxyurea in Tanzania. 

##### Lack of Infrastructure and Resources to Import or Manufacture Hydroxyurea

Manufactures viewed hydroxyurea as a special product that requires special manufacturing technology. This concern stemmed from fear of toxicity of hydroxyurea since it is also an anticancer medicine, as cited by some participants among manufacturers. One manufacturer stated the following:

“But the other thing, hydroxyurea as it is we don’t have a facility where we can manufacture because of its toxicity”.M02

Another manufacturer had nearly the same concern:

“…the systems in our factories have no enough facilities for anticancer because anticancer needs some specialized type of equipment and expertise, so those are some of the barriers which have made hydroxyurea not to be produced in our factory”.M03

##### Inadequate Knowledge on Orphan Diseases, Medicines, and the Tanzania Orphan Drug Regulation of 2018

Despite the existence of the regulation, which was meant to offset investment cost, this study found that participants among manufactures and importers are not aware of the existence and purpose of the regulation. One of the manufactures was wondering about the existence of such document and had this to say: “Regulation of? Regulation of TMDA about an orphan, no I have not made a follow-up” M04. Others acknowledged the existence of the regulation; however, they never read it deeply to understand the benefits that would otherwise be harnessed. 

“I heard (about TMDA Orphan Drug Regulation of 2018) but I have never dug deeply to understand what does it (TMDA Orphan Drug Regulation of 2018) tell, …”.M02

A similar trend was observed as regards importers, which is revealed by the following quote: “But I don’t know much about the detail of orphan drugs if I get that information then definitely, I will think of that”. I01

This was also manifested by the range of the definitions of orphan medicines, whereby some said these are drugs those are under government sponsorship for patients to get them for free or at a reduced price. Regulatory officers were also aware of the limited awareness and pointed out that the short period of time (three years) since the regulation launched might explain the inadequate awareness.

#### 3.2.2. Opportunities for Importation and Manufacturing of Hydroxyurea

This theme was abstracted from four sub-themes, namely basic knowledge on sickle cell disease and hydroxyurea, awareness of activities performed by medicines regulatory authority, sense of corporate responsibility and readiness to import or manufacture hydroxyurea, and availability of favorable government policy.

##### Basic Knowledge of Sickle Cell Disease and Hydroxyurea

Participants from importers, manufacturers, and medicines regulatory authority were able to give the meaning of SCD in terms of cause or complications. Some were too general, such as a disease of blood, and few went deep into the genetic cause and morphological shape of red blood cells, e.g., a disease where red blood cells are elongated and sickle-shaped. Furthermore, some participants were able to air out major complications such as pain crisis and low hemoglobin levels. One participant from the regulatory authority was able to give a detailed meaning of SCD, as noted in the quote below:

“…to me what I know sickle cell disease is a disease of blood, genetic disease, what I know important about sickle cell are those crises which are experienced by many sickle cell disease patients those painful crises, they have a high risk of contracting pneumonia and necessitate frequent visits (to the hospital), indeed it is a disease which troubles a lot of children born with it”. RO02

Concerning hydroxyurea use, some participants were knowledgeable about its use in managing sickle cell anemia, including the management of cancer. Furthermore, participants were able to provide their views on the benefits of hydroxyurea in the management of patients with SCD:

“…hydroxyurea help to remove those pains in peripheral areas”.I03

Another participant said the following:

“…the best one with hydroxyurea it increases hemoglobin, increases blood cell, you see all these things”.M01

Others thought hydroxyurea saves lives and that hydroxyurea is the best, and one participant confessed to being out of touch concerning hydroxyurea: “I don’t know the benefit of hydroxyurea”. 

Few participants among manufacturers, importers, and regulatory authority were able to demonstrate their knowledge on the physical-chemical properties of hydroxyurea, available formulation, availability, and price. One manufacture said, “it is hygroscopic so it is like penicillin and prone to molds infestation” M01. One participant from the importer’s group said that it is available in capsule formulation of 500 mg strength. All participants from importers and manufacturers reported that their companies are not importing or manufacturing hydroxyurea. Participants from the regulatory authority had experience with one registered hydroxyurea brand. 

Furthermore, fair knowledge expressed from participants was perceived as a strength for the prospect of importation or manufacturing of hydroxyurea.

##### Awareness of Activities Performed by the Medicines Regulatory Authority

Some participants were able to describe activities performed by TMDA in the importation and manufacturing of medicines. In addition, participants appreciated TMDA and perceived it as a support rather than an obstacle. One participant from manufacturers stated the following:

“…the authority,…TMDA so far is cooperative in terms of providing training, but also it make sure all inspections are conducted yearly and any time, when necessary, but the very scheduled inspection is conducted once every year, therefore, it ensure those Good Manufacturing Practice standards are met every time, in general, TMDA has never been an obstacle rather than a support to us”.M04

Another manufacturer expressed the following:

“So I wouldn’t worry much about TMDA”.M01

Similarly, some importers explained the procedures of registration of a product for importation. Additionally, regulatory officers explained the procedures for the development of regulations and other guidelines, especially the issue of involving stakeholders during the process. Example of quotes from one regulatory officer include the following:

“Okay, when we are developing regulations before it has been approved one of the stages is to invite stakeholders and let them know that we are developing this regulation”.RO01

Furthermore, it was noted that some participants were aware of the existence of the TMDA’s Orphan Drug Regulation of 2018. Some of the emerged ideas about reasons for the TMDA’s Orphan Drug Regulation of 2018 include “for rare diseases”, “diseases which are neglected”, and “medications which are rarely used”. Concerning the incentives offered, many talked about the waiver of product registration and retention fees. For example, one participant from importers stated the following:

“Yeah, I know because I think one time there was a meeting with TMDA and we discussed orphan medicine and through orphan medicine regulation, we can import those products which have limited consumption”.IO1

Therefore, the awareness reported and appreciation for the medicines regulatory authority are obvious factors that may act as a strength and opportunity that may also facilitate the importation and manufacturing of hydroxyurea.

##### Sense of Corporate Social Responsibility and Readiness to Import or Manufacture Hydroxyurea

Participants from the importer and manufacturer expressed their willingness to save the public interest. Codes such as the focus on “public interest” and “ready to serve” emerged during coding. An example of a quote from one manufacturer participant includes the following:

“I am quite happy to help, I have no problem at all because it is my responsibility as you know a pharmaceutical company in the country we should contribute, I don’t mind I can do it”. M01

Importers were readily willing to offer room for importation under special permits whenever the need arises. Example one was stated as follows: 

“If there is a need and we have the permits we can import, we are a registered company, the company has no problem on the issue of application to TMDA for a permit, we contact and request proforma from a specific company which own a brand of hydroxyurea then we handle the issue as a matter of public interest”. IM04

These responses from manufactures and importers show commitment to social goods and express the potential of facilitating the manufacturing and importation of hydroxyurea. 

##### Favorable Government Policy 

The study identified existing government policies that are enjoyed by manufacturers and importers and provide a potential environment that may benefit the case for hydroxyurea. Exemption of importation duty for raw materials used in pharmaceutical manufacturing was one of the policies that was hailed by one participant, who is quoted as follows: “OK we thank God that some raw materials have been exempted by the ministry of finance…”. M04 

Another policy that was noted is the special permit to import products that are currently not registered in this country. All importers were aware of this opportunity and were willing to use it in case there is an instantaneous demand for hydroxyurea. One participant said, “…if the demand will come, then we can think of importing this product under special permission”. I01

## 4. Discussion

Our study is the first of its kind to decipher the barriers for the manufacturing and/or importation of hydroxyurea in Tanzania. Furthermore, our study provides insights on the strengths and opportunities that may facilitate the manufacturing and/or importation of hydroxyurea in Tanzania. This study was able to analyze for within-case and across-case patterns that yielded explanations for the findings as documented in the subsequent sections. 

### 4.1. Limited Market and Investment Viability 

All participants expressed concern on the market size for hydroxyurea. The concern stemmed from the fact that the product is intended for a rare disease. Manufacturers and importers equally considered hydroxyurea a slow-moving product with low turnover to realize the investment made. They pointed out sources of loss, which included concern regarding the product staying for a long time in the store, which will increase risk of expiration. In addition, once the product has expired, the disposal charges will be covered by company [21]. This finding has similarly been reported by different studies that studied the investment prioritization among pharmaceutical companies [22]. The trend of disproportionate investment by pharmaceutical companies was the reason for the orphan drug movement in 1960s [15]. 

Contrary to the fear of a limited market, as narrated by many participants, available statistics on the incidence rate of 11,000 newborns with SCD every year in Tanzania [2,23] mean that the disease is rare, yet it affects a substantial number of children in Tanzania. Further, earlier initiation of hydroxyurea as per current standards of care [10,11,13,14,15,24] means there is an uninterrupted need for hydroxyurea among patients with SCD. Furthermore, a study by Costa et al. (2021) reported the investment viability of compounding hydroxyurea in Tanzania [25]. In addition, the concern regarding lack of infrastructure and resources was also refuted by Costa et al. (2021), which showed that Tanzania has the manpower and technological capabilities to manufacture hydroxyurea [25]. Therefore, the best way forward would be for SCD researchers to continue fostering collaborations with key stakeholders, specifically the ministry responsible for health, health financing agencies, pharmaceutical manufacturers, and importers. Such collaborations should aim at conducting more research and dissemination of findings to address the reported knowledge gap on disease burden and hydroxyurea use among pharmaceutical importers and manufacturers. An increase in level of awareness on the burden of SCD and use of hydroxyurea among pharmaceutical importers and manufactures might help to reduce the fear of a limited market and investment viability. 

### 4.2. Ability of Manufacturers and Importers to Harness Existing Offers to Offset Investment Costs

The inception of this study was inspired by the presence of the Tanzania Orphan Drug Regulation of 2018 [17]. This regulation was meant to be an opportunity for manufacturers and importers of pharmaceutical products targeting rare, life-threatening diseases [17]. The fact that SCD is a rare, life-threatening disease has made it an orphan disease in some countries, and hydroxyurea has been approved as an orphan medicine [16,26]. This has not been the case in our setting despite the existence of the TMDA’s Orphan Drug Regulation of 2018. This study has provided a deep insight into the problem, which, based on interviewee responses, can be divided into two spheres of reasoning.

Firstly, the knowledge gap relating to SCD and hydroxyurea was evident. Some study participants who were aware of the existence of the TMDA’s Orphan Drug Regulation of 2018 complained about lack of data on the burden of SCD as a reason for them not to exploit the offer. For one to apply for designation of a product as an orphan medicine, one has to prove that the product is for a disease that affect less than 25,000 people in Tanzania or, for life-threatening disease, has limited alternative treatments or a medicine that requires heavy investment [17]. Manufacturers and importers wondered how they would establish this kind of evidence. In addition, regulatory officers have also noted a similar problem facing clients applying for orphan drug designation. Owing to a well-established burden of SCD in Tanzania from Makani et al. (2011), (2015), Tluway and Makani (2017), and Ambrose et al. (2018) [2,3,23,27], the findings from this study show a heightened need for an information search on the part of the manufacturers and importers on one hand and researchers’ enhanced local dissemination of research findings to broader stakeholders on the other. 

Secondly, the spectrum of the level of awareness on the existence and reason for the TMDA’s Orphan Drug Regulation of 2018 spanned between partial awareness to no awareness. Moreover, the fact that stakeholders were invited only once during the development of the regulation might explain the findings. Inadequate awareness on regulation has been documented as a barrier for implementation [26]. Therefore, it is time now for the TMDA to devise a mechanism to increase awareness of the Orphan Drug Regulation of 2018. This will help in the realization of the reason behind the development of this regulation. 

### 4.3. The Need for More Incentives

Some manufacturers and importers described the scope of the orphan medicines, which went beyond what is prescribed in the TMDA’s Orphan Drug Regulation of 2018 [17]. This can further be evidenced by the recommendations expressed by the study participants, many of whom asked for the government to provide support in terms of subsidization or material support. Another manufacturer asked for partnership with the World Health Organization (WHO) or UNICEF, asking for support in funding and building the capacity of the manufacturing of hydroxyurea in Tanzania. These responses show that the current incentives offered in the TMDA’s Orphan Drug Regulation of 2018, which are the expedited registration process, waiver of registration, and retention fees [17], are perceived as not enough to offset the fear of a limited market and investment viability for orphan medicines in Tanzania. In other countries where orphan regulation exists, governments have set mechanisms for offering research and development funds to manufacturers of orphan drugs [22]. In addition, there are well-set mechanisms for reimbursement to market authorization holders of designated and approved orphan medicines [28]. These mechanisms help to cover the heavy investments during the research and development phase and offset any risk of loss as a result of slow movement of the product in the market [16]. Therefore, it is time for the government of Tanzania, in collaboration with research institutions, health financing agencies, SCD communities, pharmaceutical importers, and manufacturers, to conduct more research to gain detailed insight on the socioeconomic burden of SCD. The findings will help to make informed decisions regarding how to improve incentives so as to increase the willingness of importers and manufacturers to import or manufacture hydroxyurea for SCD and achieve the intended goal of equal access to medicines. To facilitate the process of the evaluation of mechanisms to improve incentives for the importation and manufacturing of hydroxyurea for SCD, the government of Tanzania can capitalize on the experience gained during the years of implementation of its malaria programs, such as the subsidized artemisinin-based combination therapy [29] and insecticide-treated mosquito nets [30]. However, precaution is needed when extrapolating the experience because malaria and SCD are different diseases in terms of prevalence and etiology. 

### 4.4. Potentials for Hydroxyurea Manufacturing or Importation in Tanzania

The study was able to identify existing potential to facilitate importation and manufacturing of hydroxyurea. Importers and manufacturers had fair knowledge on SCD and hydroxyurea. Luckily enough, some participants showed eagerness to learn more and acknowledged the role of this research in sensitization. 

In addition, the study noted that study participants had a sense of corporate social responsibility, which was expressed through their confirmation of readiness to manufacture or import hydroxyurea when requested for public interest. Flashback on success stories behind tackling neglected tropical diseases shows that corporate social responsibility has been the key driver. For instance, companies such as Merck AG have contributed enormously in the availability of ivermectin for filariasis [31]. Therefore, there is a room to analyze more feasible ways to utilize nearly similar models to increase hydroxyurea importation or manufacturing in our setting based on the already-expressed readiness from manufacturers and importers.

Furthermore, some participants from manufactures and importers had a good awareness on activities conducted by the medicines regulatory authority and some appreciated the support they receive from TMDA. This is stakeholders’ strength and opportunity for them to manufacture or import hydroxyurea. Finally, favorable government policies such as exemption of importation duty and special permit have been hailed by manufacturers and importers alike and hints towards potential for expansion of hydroxyurea availability in Tanzania. 

### 4.5. Strength and Limitation of the Study

Several measures were taken to ensure credibility, dependability, transferability, and confirmability of the research findings. These include a case study approach and purposive sampling of participants with more than three years of experience in the field of pharmaceuticals importation and manufacturing to obtain rich data. Furthermore, in-depth interviews in noiseless rooms within participants’ working environment and in the language preferred by interviewees ensured naturalistic and high-quality audio records. In addition, the use of a semi-structured interview guide, which was constantly evolving, allowed the free flowing of ideas. Furthermore, verbatim transcription plus field notes on non-verbal communication reduced context distortion. Finally, the use of thematic analysis and the involvement of multiple researchers during data analysis allowed for a multifaceted description and interpretation of the findings.

Since this is a qualitative study where the researcher is part of the tool for research, it is obvious that some degree of subjectivity is acknowledged as a limitation. The PI is a clinical pharmacist and a researcher who was a co-investigator in another research that involved SCD patients and so might have been affected by sympathy and empathy, which would have reduced the confirmability to this study. However, it was minimized by involving multiple researchers during data analysis. Moreover, the three companies that did not participate might have presented different perspectives, but to our best understanding, the study attained data-saturation level. 

### 4.6. Relevancy to Practice and Next Steps

This study has uncovered the main barriers to the manufacturing and importation of hydroxyurea for SCD in Tanzania. It has also highlighted the prospective facilitators for boosting the availability of this important drug. In addition, the findings have highlighted the need for further large studies, which will include policymakers and health financing agencies. In addition, the findings from this research will be disseminated in the form of a policy brief to policymakers, pharmaceutical manufacturers, and importers in order to encourage timely improvement of hydroxyurea availability in Tanzania. 

## 5. Conclusions

Inadequate understanding of SCD, use of hydroxyurea, and the TMDA’s Orphan Drug Regulation of 2018 are the important barriers that further aggravate the concern regarding a limited market and investment viability for hydroxyurea in Tanzania. Importers and manufacturers who were aware of the Orphan Drug Regulation failed to harness the opportunity because of lack of data on the burden of SCD. The presence of prospective facilitators, such as basic understanding of SCD and hydroxyurea, activities performed by TMDA, sense of corporate social responsibility, and favorable government policies, are the starting point towards increasing importation and/or manufacturing of hydroxyurea in Tanzania, the realization of which will require multi-sectoral collaboration between SCD patient communities, researchers, healthcare workers, regulators, policymakers, health financing agencies, importers, and manufacturers. 

## Figures and Tables

**Table 1 healthcare-10-02223-t001:** Socio-demographic characteristics of study participants.

Variables	Attribute	Number
Age (years)		30–90
Sex	Female	2
	Male	8
Education level	Degree	6
	Masters	4
Profession	Pharmacist	9
	Medical doctor	1
Organization	Pharmaceutical manufacturers	4
	Pharmaceutical importers	4
	Medicines regulatory authority	2
Position and role	Superintendent pharmacist	4
	Head industrial pharmacist	4
	Medicines registration officer	2
Experience (years)		5–56

**Table 2 healthcare-10-02223-t002:** Pattern of themes.

Codes	Sub-Theme	Theme
I do not know Never heard Do not affect people at the top Orphan diseases lack attention from government and policymakers There is no orphan disease Do not remember criteria for orphan disease A drug with high demand but not available Criteria involves population and availability of alternative medicines	* **Inadequate knowledge on orphan diseases, medicines, and Tanzania Orphan Drug Regulation of 2018** *	Barriers to production and importation of HU
Lack of facilities Need for specialized equipment and expertise Poor supporting infrastructures Manufacturer’s batch size requirement No ongoing importation No plan for importation	* **Lack of infrastructure and resources to import or manufacture hydroxyurea** *	
I do not have exact statistics on number of SCD patients Where to obtain data? They lack data I do not know Stakeholders lack knowledge on hydroxyurea (HU) Not aware of HU effect and side effect I do not know much about HU Never dispensed HU I do not know how HU is used in SCD	* **Inadequate awareness of sickle cell disease and hydroxyurea** *	
Market is not guaranteed Limited market No demand of HU Fear of getting loss High costs of importation	* **Limited market and investment viability** *	
Sickle-shaped cells Production of abnormal red blood cells Is a genetic abnormality Symptoms include severe pain SCD is a rare disease Hydroxychloroquine is used to treat SCD Pen V used in management in management of SCD HU available in solid dosage form HU is hygroscopic HU is not available	* **Basic knowledge of sickle cell disease and hydroxyurea** *	**Opportunities for importation and manufacturing of hydroxyurea**
Ready to serve the public need Ready to import HU when demand is available Focus on public interests Readiness for contract manufacturing	* **Sense of corporate social responsibility and readiness to import or manufacture hydroxyurea** *	
Regulatory authority provides training Regulatory authority conducts scheduled inspection or when needed Conduct annually inspection Expedited review for application of orphan medicines Exemption for orphan drug registration/retention Regulatory authority conducts scheduled inspection or when needed	* **Awareness of activities performed by the medicines regulatory authority** *	
Exemption on importation duty of raw materials Existence of special market for HU Existence of chances to order drug under special permit	* **Favorable government policy** *	

## Data Availability

Data are not available publicly because they contain sensitive interview information, and participants did not consent for their interviews to be shared publicly. The data are available from The Directorate of Research and Publication Muhimbili University of Health and Allied Sciences (contact via drp@muhas.ac.tz Tel.: +255-2150302-6) for researchers who have criteria to access confidential information.

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
