# Peer review of "Barriers and Facilitators of Availability of Hydroxyurea for Sickle Cell Disease in Tanzania; A Qualitative Study of Pharmaceutical Manufacturers, Importers, and Regulators"

_healthcare, 2022, doi:10.3390/healthcare10112223_

Round 1

Reviewer 1 Report

Although considered a rare diseases, SCD carries a high morbidity and mortality, and consequently a high disease burden, as well described by authors.  

In this paper the authors aim to identify barriers to importation and manufacturing of hydroxyurea and opportunities that may facilitate importation and manufacturing of hydroxyurea in Tanzania. They used a qualitative study design employing a case study approach. 

Interestingly, and probably expected, was the knowledge gap regarding the disease itself, the epidemiology of the disease in the country, the therapeutic benefit of hydroxyurea and the local regulations (TMDA Orphan Drug Regulation of 2018). Another important point was the perception by the different interviewed stakeholders, of a limited market and investment viability.   

It would be interesting to give readers background information about the Tanzania healthcare system and already existing public programs regarding sickle cell disease in the country. One missed aspect for me was also regarding the financial aspect of the importation and production of the drug. The limited market and investment viability for organ drugs was one of the major concerns. Authors state that “our recommendation is for the government to conduct more research on how the regulation can easily be adopted by stakeholders in order to achieve the intended goal of equal access to medicines.“  Are there already some program in the country with support of the ministry of health? Some stakeholders argued about funding of WHO and UNICEF, but nothing about how other diseases with similar epidemiology are approached in the country. 

Even if the knowledge gap can be overcome, funding will always be a problem. It would have been interesting to include in the interviews people not only involved in drug registration, but people involved in the establishment of public policies and their funding. 

Author of reference 20 is Costa and not Taher. Please confirm.

Reviewer 2 Report

Specific comments:

1.     Lines 43 page 2: there are a format error, leave space between “than11,000”

2.     In table 2, the code “Do not affect people at the top” appeared two times

3.     Explain the difference between codes: “Never heard about Hydroxyurea (HU)” and” I don’t know much about HU”, what the authors hoped to duce from each of these codes?

4.     Line 419, It would be engaging for the readers; the authors explain the mechanisms to manage with government and social agencies to broader stakeholders by researchers.

Reviewer 3 Report

1) Introduction must be improved. The relevant studies are missing in introduction. Statement of problem and why to conduct the study is also missing. 

2) In methodology, how many participants?? Why only experience are the criteria of inclusion? Where is exclusion criteria?

3) Why so many interviews were conducted?

4) Abstract and body is totally contradict?

5) Results need improvement. Which statistical test were used?

6) For Questionnaire, Which tool is used for validation.

7) The sentences are not interrelated.

8) Some of references are not related.

9) Sample size is too small.

Round 2

Reviewer 3 Report

No any